# Surfactant Effect on the Physicochemical Characteristics of Solid Lipid Nanoparticles Based on Pillar[5]arenes

**DOI:** 10.3390/ijms23020779

**Published:** 2022-01-11

**Authors:** Anastasia Nazarova, Luidmila Yakimova, Darya Filimonova, Ivan Stoikov

**Affiliations:** A.M. Butlerov Chemistry Institute, Kazan Federal University, 18 Kremlyovskaya Str., 420008 Kazan, Russia; filimon.darya@gmail.com

**Keywords:** pillar[5]arene, self-assembly, solid lipid nanoparticles, surfactant

## Abstract

Novel monosubstituted pillar[5]arenes containing both amide and carboxyl functional groups were synthesized. Solid lipid nanoparticles based on the synthesized macrocycles were obtained. Formation of spherical particles with an average hydrodynamic diameter of 250 nm was shown for pillar[5]arenes containing N-(amidoalkyl)amide fragments regardless of their concentration. It was established that pillar[5]arene containing N-alkylamide fragments can form spherical particles with two different sizes (88 and 223 nm) depending on its concentration. Mixed solid lipid nanoparticles based on monosubstituted pillar[5]arenes and surfactant (dodecyltrimethylammonium chloride) were obtained for the first time. The surfactant made it possible to level the effect of the macrocycle concentration. It was found that various types of aggregates are formed depending on the macrocycle/surfactant ratio. Changing the macrocycle/surfactant ratio allows to control the charge of the particles surface. This controlled property will lead to the creation of molecular-scale porous materials that selectively interact with various types of substrates, including biopolymers.

## 1. Introduction

Molecular-scale porous materials are used to recognize, separate and storage molecules, including proteins and oligopeptides in recent years [1,2,3]. The use of synthetic receptors with the extra stability, selectivity and specificity is in demand. An application of synthetic receptors will have a great impact on technologies based on molecular recognition [4,5,6,7,8,9]. Macrocyclic compounds are promising candidates for the creation of next generation of molecular-scale porous materials due to their cavity [10,11,12]. Microporous materials based on macrocyclic platforms are of interest because their pore sizes are comparable to small molecules or fragments of proteins [13,14,15,16].

To date, several generations of macrocyclic compounds have been synthesized (crown ethers, calixarenes, cucurbiturils, etc.) and they are capable of forming host-guest complexes [17,18,19,20]. Pillar[5]arenes are able to form various types of self-organizing structures depending on the used solvent: supramolecular polymers, supramolecular nodes and rotaxanes [21,22,23,24,25,26], and solid lipid nanoparticles (SLN) [27,28]. Initially, SLN are prepared from mixtures of lipids (hydrophobic or amphiphilic molecule, soluble in organic solvents) with, if needed, the presence of cosurfactants [29]. Since 2000, attention was turned to the use of supramolecular amphiphilic molecules (cyclodextrins, calixarene, calix[4]resorcinarenes, calixpirroles) to prepare SLN [30]. The use of macrocycles in the synthesis of SLN will make it possible to obtain particles with a dual effect in comparison with classical SLN based on non-macrocyclic lipids (mono-, di- and triglycerides, fatty acids, fatty alcohols) [29,30]. Water-insoluble drugs can be enclosed in the lipid frame of the particles. Porous surface consisting of ordered macrocycles can be used for selective recognition and retention of proteins or oligopeptides [31,32,33]. Surfactants are used to increase the aggregation stability of particles in the production of classic SLNs. Moreover, it was shown that the type and amount of surfactant affect the physicochemical properties of the formed SLN [34,35,36]. Surfactants have an amphiphilic structure; their polar parts are mainly oriented towards an aqueous phase, while their hydrophobic groups are directed towards the core center. The choice of surfactant molecules primarily depends on the chosen lipid, since they must be physicochemically compatible [37].

Thus, the design of molecular-scale porous materials based on synthetically available pillar[5]arenes in the SLN form is an important step in the development of a new generation of polymer materials for solving biomedicine problems. The addition of a surfactant to the SLN production process will increase the monodispersity of the system, its stability and packing density. Varying the macrocycle/surfactant ratio opens up the possibility of obtaining mixed SLN based on a macrocycle and a surfactant with the subsequent formation of lipid particles with a porous surface. This enables us to create homogeneous and heterogeneous self-assembling structures to increase the selectivity of produced material with respect to analytes.

In this regard, the aim of this work is the synthesis of SLN from monosubstituted pillar[5]arenes and surfactant effect on their physicochemical characteristics. The aggregation behavior of two different types of SLN was studied: (1) SLN based on monosubstituted pillar[5]arenes containing amide and free carboxyl functional groups, and (2) mixed SLN based on pillar[5]arenes and surfactants by varying their molar ratio. We assume that the design of these systems will make it possible to control the surface charge of the resulting particles for selective recognition and separation of biomacromolecules with different surface charges.

## 2. Results and Discussion

### 2.1. Synthesis of Monosubstituted Pillar[5]arenes Containing Amide and Carboxyl Groups

An important step in the development of a new generation of polymer materials for solving biomedicine problems and the creation of biomedical diagnostic systems is the design of SLN based on monosubstituted pillar[5]arenes. Varying the functional composition of the pillar[5]arene platform, we will be able to reveal structural regularities and predictably change the morphology and aggregation stability of nanoparticles. The functionalization of the amine/amide and carboxyl groups as polar fragments into the structure simultaneously will make it possible to assess the effect of the substituent on the self-assembly of macrocycles.

Monosubstituted pillar[5]arenes containing a terminal amino group were used as parent compounds (Figure 1). The compounds **1**–**3** were obtained according to the literature [38,39]. Succinic anhydride was used to introduce carboxyl fragments. The reaction was carried out in a tetrahydrofuran, which is due to the good solubility of the initial compounds in it and the necessity to use polar aprotic solvents in acylation reactions.

Compounds **4–6** were obtained in 79–83% yields. The structure of the synthesized compounds was characterized by the complexes of physical methods (^1^H, ^13^C and IR spectroscopy). The individuality was confirmed by measuring the melting point and TLC, and the composition-y mass spectrometry and elemental analysis.

It was shown earlier by our research group that compound **1** is prone to form supramolecular polymers in chloroform and does not participate in self-assembly in DMSO [39]. At the same time, compounds **2** and **3** form self-inclusion complexes regardless of the solvent nature due to the hydrogen bond between NH protons and the oxygen atom of the methoxyl fragment [40]. A similar trend persists in the reaction of the macrocycles **1–3** with succinic anhydride. No “knocking out” of amidoalkylamide fragments from the macrocyclic cavity occurs during the reaction for compounds **5** and **6** (Figure 1b and Appendix A). The absence of self-assembly in DMSO and association in chloroform were demonstrated for macrocycle **4** (Figure 1a and Appendix A).

### 2.2. Synthesis of SLN Based on Monosubstituted Pillar[5]arenes Containing Amide and Carboxyl Groups

The prepared monosubstituted pillar[5]arenes **4**–**6** were used for the subsequent synthesis of SLN by nanoprecipitation in a THF-water solution. The influence of the linker length and the concentration of monosubstituted pillar[5]arenes on the size (d, nm) and stability (PDI; ζ, mV) of the resulting particles was studied. SLN were prepared by dissolving 3 mg, 1 mg, or 0.5 mg of the corresponding pillar[5]arene in 1 mL THF (Table 1) according to the previously proposed method [27,28].

Analysis of the obtained data showed that for macrocycles **5** and **6**, containing an *N*-(amido)alkylamide fragment and forming self-inclusion complexes, the size of formed SLN-5 and SLN-6 particles is practically independent of the initial concentration. At the same time, it was found that various types of associates of submicron size with the high polydispersity indexes are formed by pillar[5]arene **4** fragment at C = 3 × 10^−4^ M.

Transmission electron microscopy confirmed the formation of SLN by macrocycles **4**–**6**. This method makes it possible to unambiguously determine the size and shape of solid aggregates. All synthesized particles form nanosized spherical aggregates according to TEM images (Figure 2). These data are in good agreement with the data obtained by the DLS method, where the average hydrodynamic particle diameter is 230 nm.

### 2.3. Surfactant Effect on the Synthesis of Mixed SLN

The stability and the possibility of creating associates are influenced by such factors as the type of the terminal group of the macrocycle and its distance from the platform [41,42,43,44]. In recent years, researchers have used surfactants to increase the stability of macrocyclic associates [45,46,47]. It was also proposed to use a surfactant to level the effect of the macrocycle concentration and obtain monodisperse stable systems. Dodecyltrimethylammonium chloride (DTAC) was used as a surfactant in our study. Its choice is due to two factors: (1) DTAC can form a bond with the terminal fragment (carboxyl group) of monosubstituted pillar[5]arenes **4**–**6** through electrostatic interactions; (2) DTAC has hydrophobic part that is tended to hydrophilic-hydrophobic interaction with the macrocyclic cavity of pillar[5]arene. Three different macrocycle/surfactant ratios were chosen to obtain mixed SLN (1:1; 1:100 and 1:1000). The effect of the surfactant amount on SLN-4 (Table 2) was studied in two concentrations (C = 3 × 10^−4^ M and C = 1 × 10^−4^ M), for which the formation of poly- and monodisperse systems was shown.

Comparing Table 1 and Table 2, it is clearly seen that there are significant decreases in the size (231 nm) and the polydispersity index (0.11) of particles for the polydisperse system, which is formed by SLN-4 at C = 3 × 10^−4^ M in an equimolar ratio with the surfactant. The negative zeta potential (−22.1 mV) indicates the presence of charged carboxyl groups (COOH) on the SLN surface. The formation of polydisperse systems and a significant PDI increasing were established at 100- and 1000-fold excess of surfactant. An increase in the particle size more than threefold to 278 nm and in polydispersity index to 0.15 were shown for SLN-4 at C = 1 × 10^−4^ M and an equimolar amount of surfactant. The value of the ζ-potential practically does not change (−30.3 mV). Close sizes of mixed particles formed by SLN-4 at two different concentrations (C = 3 × 10^−4^ M and C = 1 × 10^−4^ M) with an equimolar amount of DTAC are probably associated with the formation of aggregates similar in structure (host–guest complexes between pillar[5]arene **4** and DTAC). The addition of an excess of surfactant to SLN-4 (C = 1 × 10^−4^ M) also leads to the enlargement of the formed aggregates and a significant increase in PDI. Thus, it was shown that regardless of the initial concentration of pillar[5]arene **4**, the addition of an equimolar amount of surfactant leads to the formation of stable aggregates due to the formation of host–guest complexes between the macrocycle and the surfactant.

One- and two-dimensional NMR spectra were recorded for the pillar[5]arene **4**, DTAC and their mixture to confirm the formation of host–guest complex between them (Figure 3). The study of interaction with DTAC was carried out in methanol.

Analysis of the ^1^H NMR spectra of compound **4** with DTAC showed that the main changes occur in the chemical shifts of the surfactant molecule. The proton signals of three methyl fragments (H^a^) are shifted to the upfield by 0.1 ppm. The H^b^ and H^c^ proton signals of methylene fragments are also shifted upfield by 0.7 and 0.15 ppm respectively. There are no changes in the signals of the H^d^ and H^e^ protons. The shift of the H^b^ and H^c^ proton signals in the upfield unambiguously indicates their shielding by the macrocyclic cavity of the pillar[5]arene **4**. The change in chemical shifts of the H^a^ proton signals is due to electrostatic interaction with the carboxyl group (Figure 4).

Two-dimensional ^1^H-^1^H NOESY NMR spectroscopy was used to determine the structure of the inclusion complex of macrocycle **4** and DTAC (Figure 4). Cross-peaks between proton signals of aromatic fragments and signals of methylene protons related to quaternary nitrogen atom in surfactant molecule (H^1^/H^b^, H^1^/H^c^) are observed in the two-dimensional NMR spectrum (Figure 4). There are strong correlations between signals of methyl protons H^a^ of DTAC and signals of aromatic fragments of macrocycle (H^1^) in spectrum, as well as signals of oxymethylene protons of propyl substituent (H^4^). The presence of these cross-peaks accurately reflects the formation of the inclusion complex between pillar[5]arene **4** and the surfactant (Figure 4). It is also worth noting that cross-peaks between proton signals of methylene fragments (H^d^) of DTAC and methoxyl fragments of macrocycle (H^3^) are observed in the spectrum (Figure 4). In summary, inclusion of the surfactant molecule into macrocyclic cavity occurs in such way that a “head-to-head” complex is formed.

### 2.4. Synthesis of Mixed SLN Based on Monosubstituted Pillar[5]arenes and DTAC

The next step of investigation is the preparation of mixed SLN based on monosubstituted pillar[5]arenes **5** and **6** with an initial macrocycle concentration 1 × 10^−4^ M (Table 3). Comparing Table 1 and Table 3, it is clearly seen that the addition of an equimolar amount of DTAC does not lead to a significant change in the sizes of aggregates in the case of SLN-5. The value of the zeta potential decreases to −43.5 mV, which indicates stability increasing of the formed particles. In the case of a 100-fold excess of surfactant, a twofold increase occurs in both the size of the aggregates and the PDI of the system. This clearly indicates the formation of a mixture of the associates. An increasing of DTAC concentration (1000-fold excess) led to formation of stable mixed SLN (PDI = 0.18; ζ = 19.5 mV) which is probably due to the micelle formation of surfactant. The CMC value of DTAC (CMC = 20 mmol/L) is also confirms the micelle formation. The change in the zeta potential sign of mixed SLN to the opposite is also due to micelle formation. 

A polydispersity system is formed in the case of a 1000-fold excess of surfactant to SLN-6. Obviously, the elongation of the linker and, as a consequence, the increased conformational mobility of the tail leads to the destabilization of aggregates and dynamic equilibrium between different types of associates. The formation of stable aggregates is observed in the case of equimolar and 100-fold excess of DTAC as indicated by a significant decrease the zeta-potential (−47.4 and −32.5 mV respectively) of system (Table 3). It should be noted that aggregates become larger in the case of an equimolar macrocycle/surfactant ratio. This is probably related to the larger size of the substituent in pillar[5]arene **6**, which confirms our assumption about the influence of the tail length. An addition of a 100-fold excess of DTAC results in decrease of the formed particles size.

^1^H NMR spectra for mixtures of the compounds **5**, **6** with DTAC were recorded to establish the structure of associates, which were formed during the preparation of mixed SLN from macrocycles **5** and **6** with the surfactant (Appendix A). It was shown that for the both monosubstituted pillar[5]arenes **5** and **6** containing *N*-(amidoalkyl)amide fragment, there are no changes in the chemical shifts of the proton signals for the analyzed compound mixtures (Appendix A). The data obtained indicate that the surfactant molecule is not included in the macrocyclic cavity of the pillar[5]arenes. This is probably due to the formation of self-inclusion complexes by macrocycles **5** and **6** (Figure 1).

The ability to form solid lipid nanoparticles by macrocyles **5** and **6** changes dramatically as it expected based on their structure. This is due to a seemingly insignificant increase in the bridge between two amide fragments (from four to six carbon atoms). We assumed that all four fragments are inside the cavity in the case of the pillar[5]arene **5** and, the succinic acid residue is as close as possible to the macrocyclic fragment. Fragment of succinic acid is removed from macrocyclic cavity, and this can lead to the formation of another type of associates for macrocycle **6**. Table 3 clearly shows that at a 1:1 ratio macrocycles **5** and **6** behave similarly to each other. The picture changes dramatically in the case of 100- and 1000-fold excess. Thus, the addition of an equimolar amount of surfactant leads to the formation of stable aggregates, which is explained by the formation of host–guest complexes between pillar[5]arene and DTAC for all obtained mixed systems regardless of the concentration of initial macrocycle (Figure 5). The addition of a 100-fold excess of surfactant led to formation of polydisperse systems in the case of macrocycles **4** and **5** and monodisperse stable aggregates for pillar[5]arene **6** (Figure 5). Polydisperse systems are formed in the case of a 1000-fold excess of surfactants with macrocycles **4** and **6**. Pillar[5]arene **5** with a 1000-fold excess of DTAC forms stable aggregates, which is associated with the micelle formation of the latter (Figure 5).

The obtained mixed SLN were investigated by TEM (Figure 6). All mixed SLN form spherical nanosized aggregates according to TEM images (Figure 6). The sizes of mixed SLN are in good agreement with the DLS data.

Individual spherical particles are observed for all mixed SLN with a macrocycle/surfactant ratio of 1:1 according to TEM images. The film formation on top of the particles and partial darkening of areas are observed (Figure 6d,f) in addition to spherical aggregates in systems with an excess of DTAC. This is due to the high concentration of surfactants in the samples. It is also worth noting the homogeneity of the formed particles in the TEM images, which additionally confirms the formation of precisely SLN, and the absence of associates (micelles, liposomes) [48,49]. Thus, the addition of surfactants in the SLN synthesis allows to obtain stable monodisperse systems regardless of the linker length and the number of amide (one or two) fragments in the substituent structure of the pillar[5]arenes **4**–**6**.

## 3. Materials and Methods

### 3.1. General

All chemicals were purchased from Acros (Fair Lawn, NJ, USA), and most of them were used as received without additional purification. Organic solvents were purified by standard procedures. ^1^H NMR and ^13^C NMR spectra were obtained on a Bruker Avance-400 spectrometer (Bruker Corp., Billerica, MA, USA) (^13^C{^1^H} 100 MHz and ^1^H 400 MHz). Chemical shifts were determined against the signals of residual protons of deuterated solvent (CDCl_3_, MeOD-*d_4_*, DMSO-*d_6_*). The concentrations of the compounds were equal to 3–5% in all the records. The FTIR ATR spectra were recorded on the Spectrum 400 FT-IR spectrometer (Perkin Elmer Inc, Waltham, MA, USA) with a Diamond KRS-5 attenuated total internal reflectance attachment (resolution 0.5 cm^−1^, accumulation of 64 scans, recording time 16 s in the wavelength range 400–4000 cm^−1^). Mass spectra were obtained on a Bruker Ultraflex III MALDI-TOF instrument (Bruker Daltonik GmbH, Bremen, Germany) with *p*-nitroaniline as the matrix. Elemental analysis was performed on Perkin–Elmer 2400 Series II instruments (Perkin Elmer, Waltham, MA, USA). Melting points were determined using Boetius Block apparatus (VEB Kombinat Nagema, Radebeul, Germany).

**4-(3-Aminopropoxy)-8,14,18,23,26,28,31,32,35-nonamethoxypillar[5]arene (1)** was synthesized according to the literature [39].

**[(*N*-{4′-Aminobutyl}-amino)-carbamoylmethoxy]-8,14,18,23,26,28,31,32,35-nonamethoxypillar[5]arene (2)** and **[(*N*-{6′-aminohexyl}-amino)-carbamoylmethoxy]-8,14,18,23,26,28,31,32,35-nonamethoxypillar[5]arene (3)** were synthesized according to the literature [38].


**General procedure for the synthesis of compounds 4–6**


In a round-bottom flask equipped with a magnetic stirrer, 0.20 mmol of compound **4** (**5**, **6**) was dissolved in 8 mL of tetrahydrofuran. Then, succinic anhydride (0.09 g, 0.90 mmol) was added and the reaction mixture was refluxed for 20 h. Then the solvent was removed under reduced pressure. The residue was recrystallized from ethanol as white powder. The formed precipitate was filtered off and dried under reduced pressure over phosphorus pentoxide.

**4-(3-(3-carboxypropanamido)propoxy)-8,14,18,23,26,28,31,32,35-nonamethoxypillar[5]arene (4)**. Yield: 0.18 g (83%). M.P. = 111 °C. ^1^H NMR (DMSO-*d_6_*, 400 MHz, 298 K) δ_H_, ppm, *J*/Hz: 1.87 (m, 2H, -OCH_2_CH_2_CH_2_-), 2.28–2.31 (m, 2H, -C(O)CH_2_CH_2_-), 2.41–2.44 (m, 2H, -C(O)CH_2_CH_2_-), 3.29 (m, 2H, -OCH_2_CH_2_CH_2_-), 3.63–3.68 (m, 37H, -CH_2_- and -OCH_3_), 3.84 (m, 2H, -OCH_2_CH_2_CH_2_-), 6.75–6.85 (m, 10H, ArH), 8.00 (t, 1H, -CH_2_CH_2_CH_2_NH-, *^3^J_HH_* = 4.8 Hz), 12.13 (m, 1H, -COOH). ^13^C NMR (CDCl_3_, 100 MHz, 298 K) δ_C_, ppm: 28.82, 28.94, 29.00, 29.25, 29.37, 30.09, 35.66, 55.33, 55.42, 55.46, 55.49, 65.22, 113.34, 127.43, 127.47, 127.50, 127.53, 127.56, 149.20, 149.89, 149.98, 170.96, 173.87. IR (ν, cm^−1^): 3318 (NH), 2933, 2828, 1730, 1710 (COOH), 1673, 1648, 1612 (C(O)–NH). MS (MALDI-TOF): calculated [M+H]^+^ *m*/*z* = 894.4, [M+K]^+^ *m*/*z* = 932.4, found [M+H]^+^ *m*/*z* = 894.7, [M+K]^+^ *m*/*z* = 932.6. El. an. calcd for C_51_H_59_NO_13_: C 68.52, H 6.65, N 1.57. Found: C 68.87, H 6.38, N 1.55.

**4-[4-(3-carboxypropanamido)-butylcarbamoylmethoxy]-8,14,18,23,26,28,31,32,35-nonamethoxy-pillar[5]arene (5)**. Yield: 0.18 g (81%). M.P. = 209 °C. ^1^H NMR (CDCl_3_, 400 MHz, 298 K) δ_H_, ppm, *J*/Hz: −2.17 (m, 2H, -NHCH_2_-), −1.64 (m, 2H, -C(O)NHCH_2_CH_2_-), 0.87 (m, 2H, -C(O)NHCH_2_CH_2_CH_2_-), 1.48 (m, 2H, C(O)NHCH_2_CH_2_CH_2_CH_2_-), 2.54–2.56 (m, 2H, -C(O)CH_2_CH_2_), 2.68–2.71 (m, 2H, -C(O)CH_2_CH_2_), 3.71–3.80 (m, 37H, -CH_2_- and -OCH_3_), 4.44 (m, 1H, -CH_2_NHC(O)), 4.57 (s, 2H, ArOCH_2_-), 4.85 (m, 1H, C(O)NH), 6.69–7.02 (m, 10H, ArH). ^13^C NMR (CDCl_3_, 100 MHz, 298 K) δ_C_, ppm: 28.77, 28.97, 29.14, 29.97, 38.45, 55.25, 55.34, 55.41, 55.49, 113.24, 113.36, 127.47, 127.50, 127.59, 149.81, 149.89, 149.95, 150.00, 173.53, 173.79, 173.81. IR (ν, cm^−1^): 3379 (NH), 2936, 2830, 1730, 1702 (COOH), 1678, 1636 (C(O)–NH). MS (MALDI-TOF): calculated [M+H]^+^ *m*/*z* = 965.4, [M+Na]^+^ *m*/*z* = 987.4, [M+K]^+^ *m*/*z* = 1003.4, found [M+H]^+^ *m*/*z* = 965.9, [M+Na]^+^
*m*/*z* = 987.9, [M+K]^+^ *m*/*z* = 1003.9. El. an. calcd for C_54_H_64_N_2_O_14_: C 67.20, H 6.68, N 2.90. Found: C 67.29, H 6.65, N 2.87.

**4-[6-(3-carboxypropanamido)-hexylcarbamoylmethoxy]-8,14,18,23,26,28,31,32,35-nonamethoxypillar[5]arene (6)**. Yield: 0.17 g (79%). M.P. = 166 °C. ^1^H NMR (CDCl_3_, 400 MHz, 298 K) δ_H_, ppm, J/Hz: −1.87 (m, 2H, -C(O)NHCH_2_CH_2_-), -0.58 (m, 4H, C(O)NHCH_2_CH_2_CH_2_CH_2_-), 2.00 (m, 2H, -CH_2_CH_2_CH_2_NHC(O)-), 2.90 (m, 2H, -CH_2_CH_2_CH_2_NHC(O)-), 2.65–2.68 (m, 2H, -C(O)CH_2_CH_2_), 2.76–2.78 (m, 2H, -C(O)CH_2_CH_2_), 3.73–3.77 (m, 37H, -CH_2_- and -OCH_3_), 4.60 (s, 2H, ArOCH_2_-), 5.44 (m, 1H, -CH_2_NHC(O)), 6.02 (m, 1H, C(O)NH), 6.77–7.00 (m, 10H, ArH). ^13^C NMR (CDCl_3_, 100 MHz, 298 K) δ_C_, ppm: 26.13, 26.21, 28.78, 28.98, 29.13, 29.15, 29.20, 30.00, 54.94, 55.12, 55.36, 55.42, 55.43, 112.35, 112.86, 113.31, 127.19, 127.29, 127.44, 127.45, 149.84, 149.87, 149.93, 173.64, 174.02. IR (ν, cm^−1^): 3404 (NH), 2936, 2854, 2830, 1731 (COOH), 1676, 1645, 1533 (C(O)–NH). MS (MALDI-TOF): calculated [M+H]^+^ *m*/*z* = 993.5, [M+Na]^+^ *m*/*z* = 1015.5, [M+K]^+^ *m*/*z* = 1031.5., found [M+H]^+^ *m*/*z* = 993.6, [M+Na]^+^ *m*/*z* = 1015.6, [M+K]^+^ *m*/*z* = 1031.5. El. an. calcd for C_56_H_68_N_2_O_14_: C 67.72, H 6.90, N 2.82. Found: C 67.55, H 6.83, N 2.79.

### 3.2. SLN Preparation

The SLN suspensions were prepared by dissolving 0.5, 1 or 3 mg of **4**/**5**/**6** with/without DTAC in 1 mL of THF. After 5 min stirring, 10 mL of ultrapure water was added and the solution was stirred for one more minute. The tetrahydrofuran was subsequently evaporated under reduced pressure at 40 °C. The remaining solution was adjusted to 10 mL with ultrapure water.

### 3.3. Dynamic Light Scattering (DLS)

#### 3.3.1. Particles’ Size

The Zetasizer Nano ZS instrument (Worcestershire, UK) equipped with the 4 mW He-Ne laser (633 nm) was used for the determination of particle size. Measurements were performed at a detection angle of 173° and the software automatically determined the measurement position within the quartz cuvette. Processing of the results was performed by the DTS program (Dispersion Technology Software 4.20). The solutions were prepared using deionized water with resistivity >18.0 MΩ cm. Deionized water was obtained using a Millipore-Q purification system. In the course of the experiment, the concentrations of **4**–**6** for SLN preparation were 3 × 10^−4^, 1 × 10^−4^ and 5 × 10^−5^ M. To study the aggregation of mixed SLN, THF solutions of compounds **4**–**6** (1 × 10^−4^ M) were added to water solutions of DTAC at 1:1, 1:100, and 1:1000 macrocycle/surfactant ratios. The particle sizes were measured after 1 h mixing. Measurements were determined after 24 and 178 h three times to evaluate kinetic stability.

#### 3.3.2. Zeta Potentials

Zeta (ζ) potentials were measured on a Zetasizer Nano ZS from Malvern Instruments (Worcestershire, UK). Samples were prepared as for the DLS measurements and were transferred with the syringe to the disposable folded capillary cell for measurement. The zeta potentials were measured using the Malvern M3-PALS method and averaged from three measurements.

### 3.4. Transmission Electron Microscopy (TEM)

TEM measurements were made at the Interdisciplinary Center for Analytical Microscopy of the Kazan Federal University. Analysis of samples was carried out using a Hitachi HT7700 Exalens transmission electron microscope (Tokyo, Japan) with an Oxford Instruments X-Maxn 80T EDS detector working in STEM mode. Samples of SLN-4, SLN-5, SLN-6 and mixed SLN (5 × 10^−5^ M; 1 × 10^−4^ M; 3 × 10^−4^ M) were prepared similarly to those studied by the DLS method. 10 mL of the suspension was placed on a carbon-coated 3 mm copper grid and dried at room temperature using special holder for microanalysis. After drying, the grid was placed in the transmission electron microscope and analyzed at an accelerating voltage of 80 kV.

## 4. Conclusions

We have successfully synthesized new monosubstituted pillar[5]arenes containing both amide and terminal carboxylic fragments and showed that these macrocycles could assemble in the different supramolecular forms (self-inclusion complex and free form) in CDCl_3_, DMSO depending on the structure of the substituent. The process predicted the different aggregation behaviors of macrocycles during SLN formation in the THF/water system. Pillar[5]arenes containing *N*-(amidoalkyl)amide fragments form self-inclusion complexes. This allows us to synthesize stable SLN with average hydrodynamic diameter of 250 nm at all concentration range. Pillar[5]arene containing *N*-alkylamide fragments is in free form due to the shorter linker length. Therefore, the formation of a stable SLN system depends on the initial concentration of macrocycle. To ignore the effect of concentration, we used surfactant (DTAC). The addition of surfactants in the SLN synthesis at 1:1 molar ratio allows to obtain stable monodisperse systems regardless of the linker length and the number of amide (one or two) fragments in the substituent structure of the pillar[5]arenes. It was found that various types of aggregates are formed depending on the macrocycle/surfactant ratio. Changing the macrocycle/surfactant ratio allows to control the charge of the particle surface. A controlled change in the surface charge will lead to the creation of molecular-scale porous materials that selectively interact with various types of substrates, including biopolymers.

## Data Availability

The data presented in this study are available in Appendix A.

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
