# Peer review of "Surfactant Effect on the Physicochemical Characteristics of Solid Lipid Nanoparticles Based on Pillar[5]arenes"

_ijms, 2022, doi:10.3390/ijms23020779_

Round 1

Reviewer 1 Report

The paper can be accepted for publication. 

Author Response

Thank you for the appreciation of our work.

Reviewer 2 Report

Is not clear why is called to the nanoparticles produced with the different macrocycles solid lipid nanoparticles? Which was the lipid used?

How many samples were tested for the data presented in tables 1, 2 and 3? Why for PDI and zeta potential values the standard deviation is not presented?

Author Response

Is not clear why is called to the nanoparticles produced with the different macrocycles solid lipid nanoparticles? Which was the lipid used?

Answer:

Indeed, initially solid lipid nanoparticles are prepared from mixtures of lipids (hydrophobic or amphiphilic molecule, soluble in organic solvents) with, if needed, the presence of co-surfactants. Since 2000, the attention was turned to the use of supramolecular amphiphilic molecules to prepare SLNs: cyclodextrins [1. A. Dubes, H. Parrot-Lopez, P. Shahgaldian, and A.W. Coleman: J. Colloid Interface Sci., in press (2003). 2. E. Perrier, N. Terry, N. Rival, and A.W. Coleman: French Patent FR20006102 (2000)], calixarene, calix[4]resorcinarenes [P. Shahgaldian, M. Cesario, P. Goreloff, and A.W. Coleman: J. Chem. Soc. Chem. Commun. 326 (2002)]. In 2018, we first received SLN based on pillar[5]arene [Yakimova, L.S., Shurpik, D.N., Guralnik, E.G., Evtugyn V.G., Osin, Y.N., Stoikov, I.I. ChemNanoMat, 2018, 4(9), 919–923]. These macrocycles act as a lipid in synthesis of solid lipid nanoparticles.

We added several phrases in the manuscript: “Initially, SLN are prepared from mixtures of lipids (hydrophobic or amphiphilic molecule, soluble in organic solvents) with, if needed, the presence of co-surfactants [29]. Since 2000, the attention was turned to the use of supramolecular amphiphilic molecules (cyclodextrins, calixarene, calix[4]resorcinarenes, calixpirroles) to prepare SLN [30].”

How many samples were tested for the data presented in tables 1, 2 and 3? Why for PDI and zeta potential values the standard deviation is not presented?

Answer:

We carried out 3 to 6 repeated experiments. We added values the standard deviation in tables 1, 2 and 3

Reviewer 3 Report

Manuscript Title: Surfactant effect on the physicochemical characteristics of solid  lipid nanoparticles based on pillar[5]arenes

Journal Title: Int. J. Mol. Sci.

Authors: Anastasia Nazarova, Luidmila Yakimova, Darya Filimonova, and Ivan Stoikov

Manuscript ID: ijms-1537760

In this work, the authors synthesis of SLN from monosubstituted pillar[5]arenes and surfactant effect on their physicochemical characteristics. The work is a logical continuation of their previous research (for instance https://doi.org/10.3390/nano11040947, https://doi.org/10.3390/ijms22116038) and with a high impact on literature. This manuscript is suitable for publication in Int. J. Mol. Sci. However, there are major concerns to be addressed before its publication as follows:

  • S13-S.15. it is recommended to name the x-axis the wavenumber instead of the wavelength. It is recommended to represent peak wavenumbers to whole numbers rather than decimal. Some captions are superimposed, please correct.
  • S16-S42. Please add captions to figures what each colored line on the graph means.
  • Synthesized compounds contain of nitrogen. It is recommended also to performe 14N or 15N NMR study.
  • It is well known that zeta potential depends by the pH of the solution. Did authors use buffers for zeta potential mearsuments?
  • How product was purified? Did you recrystallized the product? If yes, please describe it in the experimental part.
  • Did you performed UV-Vis spectra? You mentioned it in the experimental part, however, there are no in the Results and discussion.
  • Authors should describe general procedure for the synthesis of compounds in more details.

Author Response

In this work, the authors synthesis of SLN from monosubstituted pillar[5]arenes and surfactant effect on their physicochemical characteristics. The work is a logical continuation of their previous research (for instance https://doi.org/10.3390/nano11040947, https://doi.org/10.3390/ijms22116038) and with a high impact on literature. This manuscript is suitable for publication in Int. J. Mol. Sci. However, there are major concerns to be addressed before its publication as follows: 

First and foremost, we would like to thank esteemed Reviewer for careful consideration of the manuscript. In accordance with the comments, the following changes have been made:

S13-S.15. it is recommended to name the x-axis the wavenumber instead of the wavelength. It is recommended to represent peak wavenumbers to whole numbers rather than decimal. Some captions are superimposed, please correct.

Answer:

The X-axis has been renamed to the wavenumber. All peak wavenumbers have been round to whole numbers. All captions have been corrected.

S16-S42. Please add captions to figures what each colored line on the graph means.

Answer:

The each colored line on the graph means one repeated measurement. We added the necessary sentence to figure’s capture.

Synthesized compounds contain of nitrogen. It is recommended also to performe 14N or 15N NMR study.

Answer:

The containing of the nitrogen has been confirmed for all synthesized compounds by elemental analysis. Routine methods (1H and 13C NMR) are sufficient to determine the molecule structure.

It is well known that zeta potential depends by the pH of the solution. Did authors use buffers for zeta potential mearsuments?

Answer:

In our experiments we used Milli-Q water.

How product was purified? Did you recrystallized the product? If yes, please describe it in the experimental part.

Answer:

The product has been recrystallized from ethanol. The necessary clarifications have been added in manuscript.

Did you performed UV-Vis spectra? You mentioned it in the experimental part, however, there are no in the Results and discussion.

Answer:

This is an unfortunate misunderstanding. UV-Vis spectra of synthesized compounds and its mixtures did not register.

Authors should describe general procedure for the synthesis of compounds in more details.

Answer:

The description of general procedure for the synthesis of compounds in more details has been added in the text.

Reviewer 4 Report

A very good study.  The figures are especially well-presented. 

Author Response

(The authors gave the same response as above.)

Round 2

Reviewer 3 Report

The authors have addressed all my comments/concerns in this revised manuscript. I have no additional concerns regarding this manuscript.

Manuscript could be recommended for publication in this form